# Modelling Epithelial Ovarian Cancer in Mice: Classical and Emerging Approaches

**DOI:** 10.3390/ijms21134806

**Published:** 2020-07-07

**Authors:** Razia Zakarya, Viive M. Howell, Emily K. Colvin

**Affiliations:** 1Bill Walsh Translational Cancer Research Laboratory, Kolling Institute, Northern Sydney Local Health District, St Leonards, Sydney, NSW 2065, Australia; razia.zakarya@sydney.edu.au (R.Z.); viive.howell@sydney.edu.au (V.M.H.); 2Northern Clinical School, Faculty of Medicine and Health, University of Sydney, Sydney, NSW 2006, Australia

**Keywords:** epithelial ovarian cancer, high-grade serous, genetically engineered mouse, syngeneic models

## Abstract

High-grade serous epithelial ovarian cancer (HGSC) is the most aggressive subtype of epithelial ovarian cancer. The identification of germline and somatic mutations along with genomic information unveiled by The Cancer Genome Atlas (TCGA) and other studies has laid the foundation for establishing preclinical models with high fidelity to the molecular features of HGSC. Notwithstanding such progress, the field of HGSC research still lacks a model that is both robust and widely accessible. In this review, we discuss the recent advancements and utility of HGSC genetically engineered mouse models (GEMMs) to date. Further analysis and critique on alternative approaches to modelling HGSC considers technological advancements in somatic gene editing and modelling prototypic organs, capable of tumorigenesis, on a chip.

## 1. Introduction

Epithelial ovarian cancer (OC) is the third most common gynaecologic cancer [1], with patients experiencing the worst prognosis and highest mortality rate of all gynaecological malignancies [2]. OC is three times more lethal than breast cancer [3] and mortality is projected to increase significantly [4]. OC lethality is attributed to a problematic triad of asymptomatic tumour growth, delayed onset of symptoms, and current dearth of effective screening methods that together make early tumour detection difficult [5,6]. After typical late-stage diagnosis, treatment options include tumour debulking surgery, followed by chemotherapy with taxane and platinum, which OC patients initially respond well to but the incidence of recurrence with concurrent decrease in sensitivity to chemotherapeutic options is high [7].

Preclinical mouse models are essential for investigating carcinogenesis and testing new therapies, however, have proven somewhat difficult to achieve in OC. This is partly due to OC encompassing several distinct disease subtypes, each with a unique histological and molecular pathology. The four OC subtypes which have been classified histologically are mucinous, clear cell, endometrioid, and serous. In 2004, a broad classification separating ovarian cancers to type I and type II tumours was introduced [8], prompting the division between high- and low-grade serous OC. Further genomic analyses allowed for the identification of clusters of germline and somatic mutations, revealing molecular phenotypes underlying the existing histological subtypes [8,9,10], the findings of which are outlined below in Table 1. These molecular phenotypes have been used to guide the development of genetically engineered mouse models (GEMMs) that attempt to recapitulate the initiation and development of OC. Table 1 provides a summary of GEMMs that have been developed to resemble mucinous, endometrioid and low-grade serous OC. To our knowledge, no GEMMs have been developed that resemble clear cell OC. The most effort has undoubtedly gone into the development of GEMMs for the most common and aggressive subtype of OC, high-grade serous cancer (HGSC), which will be the main focus of this review. HGSC GEMMs have been reviewed in greater detail previously [11] however a summary of the most recent models is detailed in Table 1.

## 2. High-Grade Serous Ovarian Carcinoma

HGSC accounts for the largest proportion of deaths from epithelial ovarian cancer, with over 70% of deaths being attributed to the disease [36,37]. HGSC is the most aggressive type of epithelial OC, with a median progression-free survival (PFS) rate of 18 months [7]. Much of the molecular pathophysiology of HGSC has been uncovered by The Cancer Genome Atlas (TCGA) [30], through which HGSC can be further divided into the following four subtypes based on TCGA gene expression profiles: differentiated, immunoreactive, mesenchymal, and proliferative. Interestingly, the initial TCGA cohort did not show statistically significant differences in survival time between subtypes but combining the subtypes with prognostic genetic information allowed for the prognostic Classification of Ovarian Cancer (CLOVAR) model to be developed [38]. 

Through prognostic genomics it was shown that the immunoreactive subtype had the longest survival, the proliferative subtype had significantly worse overall survival than the immunoreactive subtype but obtained the greatest benefit from bevacizumab treatment, and that mesenchymal subtype patients had significantly worse overall survival than patients of the immunoreactive subtype [39]. A study introducing a novel histopathological classification system that was closely correlated with the TCGA molecular subtypes was able to also demonstrate that patients with mesenchymal tumours may be more sensitive to paclitaxel [40]. However, it should be noted that further studies have carefully microdissected tumour and stromal tissue to demonstrate that the TCGA mesenchymal and immunoreactive subtypes are likely to be a reflection of increased contribution tumour infiltrating and surrounding stromal cells [41,42] and highlighting the potential impact of the tumour microenvironment on patient survival and response to therapy. Given that HGSC can be further divided into distinct subtypes that have distinct survival profiles and response to therapies, this may have downstream effects on future selection and use of HGSC GEMMs, particularly in the context of emerging immunotherapy and tumour microenvironment-targeting therapies. 

Interactions between tumour cells and non-malignant stromal cells – such as endothelial cells, fibroblasts, adipocytes, mature and circulating progenitor immune cells – have been shown to promote tumour initiation, growth, and metastasis using both in vitro and in vivo models [43]. The established interplay further underpins the necessity for a GEMM that accurately recapitulates HGSC; allowing for dynamic investigations on prevention, early detection, and development of new treatment strategies that consider the interplay between the tumour and its surrounding microenvironment. Such in vivo models offer dynamic insights that would not be exposed using cell lines and are limited in patient-derived xenograft models. However, there are relatively few robust GEMMs of HGSC despite our knowledge of the key driver mutations – this is predominantly due to historical contention of the cell of origin and the lack of reliable and specific promoters for the main two cell-of-origin contenders [44]. Also, due to long latency, complex breeding strategies and the associated expense, the promising GEMMs that have been developed have not extended into widespread use by ovarian cancer researchers. This review will highlight recent HGSC GEMM developments, discuss whether or not they are serving their purpose as preclinical research tools, and discuss new approaches and future directions for GEMMs in preclinical HGSC research.

## 3. GEMMs of HGSC—Summary and Updates

The path to developing a successful HGSC GEMM has been confounded by an initial lack of understanding of the cell-of-origin for HGSC. The primary consensus was that epithelial OCs, true to their name, originated in the ovarian surface epithelia (OSE) [45], however more recently it has been proposed that HGSC originates in the fimbriae of the fallopian tube (FT = oviduct in mice) [33,46,47]. The matter is further complicated by limited OSE or oviduct-specific promoters that can be used to target genetic changes to these cell types using the CRE-loxP system. Many OC GEMMs were developed in the early 2000s and these are reviewed in greater detail by Howell [11] and summarised below, with a greater focus given to the more recently developed models.

Early promoter-driven GEMMs relied on promoters that target the developing reproductive system such as anti-Müllerian hormone receptor 2 (*Amhr2/MISIIR*). As such, these models did not specifically target the OSE or oviduct, leading to the development of uterine leiomyosarcomas, granulosa cell tumours, low-grade serous ovarian tumours or HGSC depending on the genes targeted [48,49,50].

The lack of highly specific promoters for the OSE led to the development of a surgical technique by Flesken-Nikitin et al. [51] to drive genetic changes to the OSE. Briefly, the technique involves intra-bursal delivery of adenoviral CRE (ADCRE) into the space between the ovary and the ovarian bursal membrane. The method demands a high level of technical competence and consideration towards the construct used, estrous stage of the animal, solutions used, and the multiplicity of infection of the adenovirus to obtain reproducible results [11]. In addition, cells in structures surrounding the ovarian bursa, including the oviduct can also potentially be exposed to ADCRE due to proximity or through CRE leakages that may occur along the needle path [52]. Therefore, many of the models developed using this technique tended to have lower penetrance than promoter-driven models. 

More recently, promoters have been developed to target mutations to the oviductal epithelial cells. These include paired box 8 (*Pax8*) and oviductal glycoprotein 1 (*Ovgp1*). An inducible Pax8 promoter (Pax8-rtTA;TetO-Cre) was used by Perets et al. to target knockout of *Brca1/2*, *Trp53* and *Pten* in the secretory cells of the mouse oviduct [34], thereby effectively modelling HGSC originating in the fallopian tube (FT) at a 100% transformation rate when using homozygous *Brca1* or heterozygous *Brca2* mutations. Mutations to *Brca1/2* resulted in significantly lower latency (13 weeks) than *Tp53*^-/-^/*Pten*^-/-^ only mutants, with the models developing features consistent with serous tubular intra-epithelial carcinoma (STIC) lesions (secretory cell proliferation, loss of polarity, cellular atypia, and serous histology) and metastases patterns characteristic of HGSC. The success of this model is underpinned by the fact that PAX8 is a marker of secretory cells in the FT, but not the ovaries [53,54] and expressed in HGSC tumours [55,56] in humans and mice. However, the use of *Pax8* as a promotor for the development of oviduct derived HGSC GEMMs is limited by its lack of specificity as Pax8 is also highly expressed in the thyroid and kidney. Models using *Pax8* driven perturbations in *Tp53* and *T121* have resulted in enlarged thymi resulting in respiratory distress and subsequent early lethality [35].

*Ovgp1* encodes for a large epithelial glycoprotein that is secreted from non-ciliated epithelial cells and has 100-fold higher expression in the oviduct compared to other tissues [57] and represents a more specific promoter for the development of oviduct-derived models of HGSC. Zhai et al. [34] crossed an existing *Ovgp1-iCreER^T2^* transgenic line [25] with variations in alleles of *Rb1*, *Trp53*, *Brca1*, *Nf1* and *Pten*. Mice with aberrant *Brca1*, *Trp53*, *Rb1* and *Nf1* (BPRN) and mice with aberrant *Brca1*, *Trp53* and *Nf1* (BPN) mice had the highest penetrance characteristic of HGSC, with the former demonstrating shortest latency. Further characterisation of BPRN showed these mice develop tumours highly correlating in transcriptional signatures with human HGSCs of the immunoreactive and mesenchymal TCGA subtypes [58]. However, it should be noted that off-target tumours occurred in a sizeable proportion of BPRN mice and included lymphoma, thyroid, lung, skin, and breast malignancies; this is not surprising given that some mice contained germline alterations in *Brca1* and *Trp53* and the long latency to tumour development. Finally, tumour development in mice harbouring *Brca1*, *Trp53* and *Pten* aberrations (BPP mice) was the most rapid and tumours displayed phenotypes not seen in the other genotypes such as mucinous metaplasia. Interestingly, these are the same genetic aberrations Perets et al. used in their *Pax8*-driven HGSC model. However, they did not observe these features, raising the possibility that *Pax8* and *Ovgp1* are initiating carcinogenic changes in different cell populations. 

Although later GEMMs have focused on FT/oviduct-derived HGSC, emerging molecular profiling suggests that the OSE may still represent the cell of origin for a significant proportion of HGSC [59,60]. These findings have been corroborated by proteomic analysis [61], and it is predicted that the two subtypes of HGSC demarcated by cell-of-origin may respond differently to therapy [62], with OSE-derived tumours demonstrating higher resistance to chemotherapy [61]. To explicate the matter, Zhang et al. [35] targeted identical genetic aberrations (*Trp53* mutation and *Rb* inactivation) to either the oviduct or the OSE to directly compare tumours arising from different cell lineages. This study was instrumental in not only showing that HGSC can arise in both the oviduct and the OSE, but that the cell of origin impacts tumour growth, metastasis and response to therapy. Importantly, as part of this study, they managed to develop a new promoter-driven model of OSE-derived HGSC using *Lgr5*; a gene involved with OSE homeostasis and repair, with concentrated expression in the ovarian hilum in adult mice [63]. By successfully inducing HGSC in *Lgr5Cre^ERT2^* mice, Zhang et al. provide further evidence for the dual origin of HGSC tumours.

As demonstrated by the above models, GEMMs attempting to recapitulate serous epithelial OC have focused on known molecular pathognomonic markers, such as mutations or loss of *TP53*, *BRCA1/2*, and *RB*, as well as alterations in genes not commonly associated with HGSC such as *PTEN*. It becomes clear from these GEMMs that to achieve the required genetic changes to induce HGSC in mice extensive breeding programs are required, which may act as an impediment for many researchers to use as they prove to be challenging, time-consuming, and expensive. To overcome this, SV40Tag has been widely used in the development of many GEMMs of cancer, including OC [64,65,66]. It is particularly relevant in HGSC as SV40Tag leads to the inactivation of p53 and Rb. Previous studies have demonstrated that SV40Tag activation with ADCRE or via the *Amhr2* promoter can induce ovarian tumours in a relatively efficient manner [64,67,68]. However, the dearth of studies comparing this model to the molecular phenotype of human HGSC and the lack of OSE or FT specificity of the *Amhr2* promoter may provide limitations to its use as a model to recapitulate human HGSC. 

Despite recent improvements to GEMMs of HGSC their widespread use has not eventuated, we surmise that this is due to several factors – extensive breeding requirements to introduce up to 4 or more genetic alterations to the OSE or FTE, long latency to tumorigenesis (in excess of 12 months in some instances), and instances of low penetrance and metastatic spread.

## 4. Alternative Approaches and Future Directions

GEMMs offer the ideal system to recapitulate spontaneous tumour development through a process of transformation of normal cells to development of precursor lesions before formation of invasive and metastatic tumours, which is why GEMMs remain the platform of choice for investigations into HGSC prevention strategies and early carcinogenesis. However, GEMMs may not be essential to researchers wanting to test new therapies and further investigate the biology of established tumours, metastases, and tumour microenvironment. To this end, there are other approaches in use and emerging that may appeal to researchers as an alternative to GEMMs due to the models’ higher efficiency and reduced cost (summarised in Figure 1).

The tumour microenvironment is increasingly recognised to play a significant role in all cancers [69,70,71], including HGSC [72,73,74]. Therefore, a preclinical model that not only recapitulates the genomics of HGSC but also the intricacies of the tumour microenvironment is highly desirable. Considering the importance of the immune system in tumorigenesis and the increased interest in the use of immunotherapies to treat cancer, the use of human cell lines and patient-derived xenografts (PDXs) is limited and syngeneic mouse cell lines may be a better option to epitomise the interactions between the tumour and stroma.

### 4.1. Syngeneic Cell Lines

Several syngeneic ovarian cancer cell lines have been developed as models of HGSC. Tumour cell engraftment commonly occurs via intra-peritoneal (i.p.) injection in order to mimic the widespread metastatic disease seen in HGSC patients. The other major methods of engraftment include sub-cutaneous (s.c.) and sub-bursal injection. S.c. injection allows for rapid tumour development and ease in measuring tumour volume with a caliper, whilst i.p. and intrabursal injection allow for a physiologically relevant microenvironment [75]. However, models using i.p. and intrabursal injection commonly result in widespread peritoneal disease with widespread metastases [75], which while more clinically relevant can be more complex when analysing changes in tumour growth. When considering an engraftment location, it is imperative to consider the features of the ovarian cancer cell line being used as investigations have uncovered variations in cell line tumorigenicity when comparing s.c., i.p., and intrabursal xenografts [76]. Interestingly, when comparing gene expression of s.c. to i.p. tumours it was shown that there was no overlap between the s.c. and i.p. datasets and those cell lines showing preference for i.p. growth expressed genes more commonly represented in primary cancers [76]. Syngeneic ovarian cancer cell lines are most commonly injected i.p. and a summary of survival times for recipient mice is presented in Table 2.

The ID8 cell line is derived from explanted ovaries of C57Bl/6 mice, from which OSE was grown in vitro in the presence of EGF [80] and is the most widely used and recognised syngeneic mouse OC cell line. However, genomic analysis has shown that ID8 tumours are not highly representative of human HGSC, as they lack pathognomonic HGSC mutations in *Trp53, Brca1*, *Brca2*, *Nf1*, and *Rb1* [77]. The study further showed that p53 remains transcriptionally active in ID8 tumours and that ID8 cells were able to form Rad51 foci in response to DNA double strand breaks, thereby demonstrating homologous recombination (HR) competence which are both features uncharacteristic of human HGSC. Walton et al. [77] were able to utilise the CRISPR/Cas9 system to engineer two mutated ID8 cell lines, one with a single *Trp53* knockout (*Trp53*^-/-^) and another double knockout model with the addition of *Brca2*^-/-^(*Trp53*^-/-^;*Brca2*^-/-^). It was shown that the *Trp53*^-/-^ and *Trp53*^-/-^;*Brca2*^-/-^ models had significantly shorter median survival than the native ID8 orthotopic model when grown i.p. in mice, with the single knockout having the shortest median survival of the two (Table 2), demonstrating the importance of *Trp53* knockout in survival latency. Morphological features of the *Trp53*^-/-^ tumours were similar to the wild-type ID8 tumours, with the exception of increased peritoneal and diaphragmatic metastatic deposits in the former, whilst *Trp53*^-/-^;*Brca2*^-/-^ cells formed significantly less ascites than native ID8 or the single knockout mice. As with HGSC tumours, *Trp53*^-/-^;*Brca2*^-/-^ tumours did not display HR competence. Interestingly, when comparing the tumour microenvironment of the *Trp53*^-/-^ and *Trp53*^-/-^;*Brca2*^-/-^ models with the native ID8 model, each variation displayed phenotypically different immune infiltrates. Tumours derived within the single knockout model had significantly higher levels of macrophage infiltration than ID8 and *Trp53*^-/-^;*Brca2*^-/-^ tumours; whilst the double knockout model developed tumours with the presence of CD3^+^ predominant lymphoid aggregates, with CD8^+^ populations on the periphery. There was no evidence of lymphoid aggregates in the *Trp53*^-/-^ or ID8 tumours. The phenotypic variation in immune cell infiltrates highlights the importance of using an accurate syngeneic cell line to develop a reliable immunocompetent model, capable of accurately reflecting the interactions between the tumour and the microenvironment. 

A follow up study [78], again using ID8 cells, included edits to *Brca1* (*Trp53*^-/-^;*Brca1*^-/-^), *Nf1* (*Trp53*^-/-^;*Nf1*^-/-^), *Pten* (*Trp53*^-/-^;*Pten*^-/-^) and a triple knockout with *Pten* (*Trp53*^-/-^;*Brca2*^-/-^;*Pten*^-/-^). When comparing survival in mice with *Trp53*^-/-^ tumours, it was shown that there was no significant difference in survival of mice with *Trp53*^-/-^;*Brca1*^-/-^ tumours (46 days), but *Trp53*^-/-^;*Nf1*^-/-^ and *Trp53*^-/-^;*Pten*^-/-^ tumours displayed significantly shorter survival (Table 2). Tumours generated from the triple deletion (*Trp53*^-/-^;*Brca2*^-/-^;*Pten*^-/-^) had significantly shorter survival (40 days) than *Trp53*^-/-^ and *Trp53*^-/-^;*Brca2*^-/-^ tumours (57 days), but longer survival than *Trp53*^-/-^;*Pten*^-/-^. Mutants containing edits to *Brca1/2* were significantly more sensitive to PARP inhibition with rucaparib treatment in vitro than those with single mutations in *Trp53* only. There was no significant difference in rucaparib sensitivity between *Trp53*^-/-^;*Brca2*^-/-^ and *Trp53*^-/-^;*Brca2*^-/-^;*Pten*^-/-^ cells in vitro and tumours in vivo. Whilst results for cisplatin therapy mirrored those of rucaparib in vitro, in vivo cisplatin administration lead to significant differences in survival times with mice harbouring the *Trp53*^-/-^;*Brca2*^-/-^ tumours exhibiting the longest survival and *Trp53*^-/-^;*Nf1*^-/-^ and *Trp53*^-/-^;*Pten*^-/-^ tumours having the worst survival. Thereby demonstrating that tumours derived from differing mutations, will respond differently to treatment and highlighting the importance of choosing multiple syngeneic cell lines for preclinical testing of therapies. Further characterisation of these cell lines is required to determine which human HGSC subtype they most closely resemble.

Four polyclonal cell lines derived from the *Pax8-rtTA*; *TetO-Cre*; *Brca1^loxP/loxP^*; *Trp53^mut^*; *Pten^loxP/loxP^* GEMM [33] backcrossed onto a C57Bl/6 background were used to develop a syngeneic mouse model of FT-originating HGSC [79]. The study verified genomic concordance between syngeneic models derived from the polyclonal cells and human HGSC in terms of key orthologous gene copy number alterations (CNAs) and genes encoding for transcription factors. It was further shown through transcriptional network analysis that the polyclonal cell transplants lead to the development of tumours that reflect the transcriptional patterns of the human HGSC tumour microenvironment. Immune cell profiles varied between the metastases of each polyclonal cell line, with one (HGS1) showing the closest resemblance to human HGSC metastases, with immune cell analysis of tumours showing that there were significantly fewer CD3^+^ T cells in all four polyclonal cell-derived mouse tumours compared to human HGSC. Upon assessing the stroma, it was shown that the models demonstrated enhanced expression of collagens, matrisome glycoproteins, proteoglycans, ECM regulators, and ECM-secreted factors compared to omental controls. As with human HGSC tumours, those developed in the syngeneic mouse models had an increased expression of matrisome genes, comparable ECM stiffness, and a correlating increase in the Matrix Index [81]. The upregulation of six matrisome molecules (fibronectin 1, versican, collagens 1A1 and 11A1, cathepsin V, and cartilage oligometric matrix protein) is associated with disease score [81] in human HGSC; all six were detected in the aforementioned polyclonal cell-derived syngeneic models but to varying extents.

The examples listed above demonstrate how a syngeneic mouse model can be used to circumvent the extensive and time-consuming breeding programs required to establish a GEMM, by using a more cost-effective transplantable model of HGSC derived from either the OSE or FT.

### 4.2. Organoids

Organoids are a way of modelling 3D multicellular interactions in vitro and have recently been gaining more attention for cancer modelling. Briefly, organoids are grown in culture using pluripotent stem cells or organ progenitors that differentiate and self-organise to form an organ-like structure that behaves much like the archetypical organ in vivo. Such cellular organisation is feasible due to the use of a 3D cell culture matrix comprised of Matrigel, small molecules and growth factors that mimics the basement membrane. Organoid lines derived from surgically resected primary human HGSC tumours and metastatic lesions have been developed from both FT and OSE origin [82]. It should be noted that there was a high level of morphological variation in the HGSC organoids, with varying degrees of cellular organisation, and an absence of vessel and connective tissue elements. Majority of the organoids demonstrated aneuploidy and CNAs characteristic of OC. However, there were a minority subgroup that did not. Interestingly, CNAs were maintained after prolonged passage, making them attractive for in vitro use. The organoids showed conserved aberrations characteristic of HGSC, such as mutations in *TP53*, and loss of *RB1* and *PTEN* and a maintained methylation profile after passaging. It was further demonstrated that HGSC-modelling organoids can be genetically modified in vitro in a stable manner. Organoid lines can be retransplanted into an immunocompromised mouse as an orthotopic xenograft and have been shown to develop tumours that maintain high fidelity with the original tumour and organoid in bladder [38] and liver [83] cancer. Therefore, future studies should aim to develop models of HGSC by transplanting in vitro modified organoids into a xenograft model.

Another efficient manner of using organoids for HGSC modelling in vivo is similar to the syngeneic cell lines discussed above, wherein the organoid is developed directly from a GEMM, transformed in vitro and then transplanted back into recipient mice. This method has been carried out effectively by Zhang et al. [35] to develop FT- and OSE-derived organoids, by targeting *Pax8* and *Lgr5*^+^ respectively, and shown to develop tumours comparable to human HGSC tumours. The utility of this model is compounded by being able to transform organoids in vitro to direct mutations to the right cell of origin and circumvent off-target issues associated with the use of the *Pax8* and *Lgr5*^+^ promoters. Another promising feature of organoids includes the ability to form human microtissues from primary cancer cells [84].

### 4.3. Somatic Gene Editing

A common criticism of GEMMs in cancer is that many of them are developed by using promoters that are expressed embryonically, resulting in loss of tumour suppressor genes and/or expression of oncogenes early on in development rather than in the adult organ. This can be circumvented by using inducible CRE systems such as the TetOCre or tamoxifen-inducible CRE. However, as seen in the recent paper by Zhang et al. [35], at times there is inherent “leakiness” of CRE that may interfere and lead to activation of genes early and in the wrong locations. With the increase in use of gene editing technologies, such as CRISPR/Cas9, new GEMMs are being developed that involve somatic gene edits in living adult animals, which is more akin to how cancers arise in the adult. 

The CRISPR/Cas9 system is a powerful tool capable of recognising specific gene sequences and inducing a double-stranded break through non-homologous end joining (NHEJ) to induce somatic gene edits without having to undertake the onerous, time-consuming and expensive crossbreeding involved with developing a GEMM. Further, by inducing such edits somatically in adult mice, CRISPR/Cas9 allows for a more accurate representation of the sporadic mutations typical of tumorigenesis. Modifications to the Cas9 nuclease has allowed for a wide scope of applications that have been comprehensively reviewed by Anzalone et al. [85]. A significant drawback of this system, however, is the complexity involved in somatic CRISPR delivery. Direct DNA transfection has been shown in multiple tissue types, including the liver [86], pancreas [87], and brain [88] but is limited to a few organ types. To overcome these limitations, most CRISPR models to date rely on viral vectors such as adenovirus and adeno-associated viruses. Viral mediated delivery has been used to induce combinatorial alterations in a lung cancer model in vivo [89], demonstrating that the tool could be harnessed to induce the multiple pathognomonic HGSC genomic aberrations. An important consideration for using CRISPR/Cas9 systems for somatic gene editing is low in vivo editing efficiency of 1-5% or lower. Although the complete mechanisms underlying low editing efficiency remain to be elucidated, studies have shown that primary cells prefer to undergo apoptosis [90] or use NHEJ [91]. It has been postulated that this is due to wild-type p53 being activated by Cas9 mediated double strand breaks to initiate growth arrest and apoptosis [92,93]. There has yet to be a CRISPR/Cas9 HGSC model developed to date. However, given that viral delivery has previously been used to deliver CRE recombinase to the OSE/oviduct, somatically edited GEMMs of HGSC would be feasible.

### 4.4. Tumour-on-a-Chip

The tumour-on-a-chip systems are microfluidic devices that aim to recapitulate features of tumour physiology. The system substrate is commonly glass or optically clear polymers with perfused, hollow microchannels. Typically, two or more cell types are set up within these channels and the system can be set up to have the cell-to-cell interactions occurring directly or indirectly. The intricacy of this system and the addition of microfluidics adds a dimension of physiological mimicry and gives the researcher precise control over variables [94]. The simplest version of a tumor-on-a-chip system is comprised of a tumour spheroid placed in a microfluidic system, whilst more complex versions can include 3D cancer tissues in contact with non-cancerous tissues and spheroids surrounded by healthy tissues [95]. These models are bolstered by the possibility of using human derived spheroids or organoids for a more accurate recapitulation of patient tumours with lung cancer models having been used to investigate the interactions between tumour cells and CAFs [96] and drug sensitivity [97]. The tumour-on-a-chip system has been used to model HGSC using CRISPR-Cas9 edited *TP53* knockouts in dog oviductal epithelia, wherein cells lost normal morphology, exhibited increased proliferation and DNA double strand breaks [98]. Upon genetic analysis the *TP53* knockouts had a decreased expression of *PTEN* and *RB1* but no changes on *BRCA1/2*. This study is a promising first step in modelling HGSC on a chip, however further studies investigating interactions between the tumour and stroma are necessary to further validate this method.

## 5. Conclusions

Compared to other tumour types, GEMMs of HGSC have been notoriously difficult to develop. Contention surrounding the cell of origin, the lack of robust and specific promoters, extensive breeding requirements, and long latency to tumour development have all contributed to these difficulties. Despite the development of several models that closely resemble HGSC, the aforementioned challenges have meant that these GEMMs have not been widely adopted for preclinical research. New technologies are rapidly emerging that will lead to the development of better GEMMs or provide strong alternatives for their use in preclinical research.

## Figures and Tables

**Figure 1 ijms-21-04806-f001:**
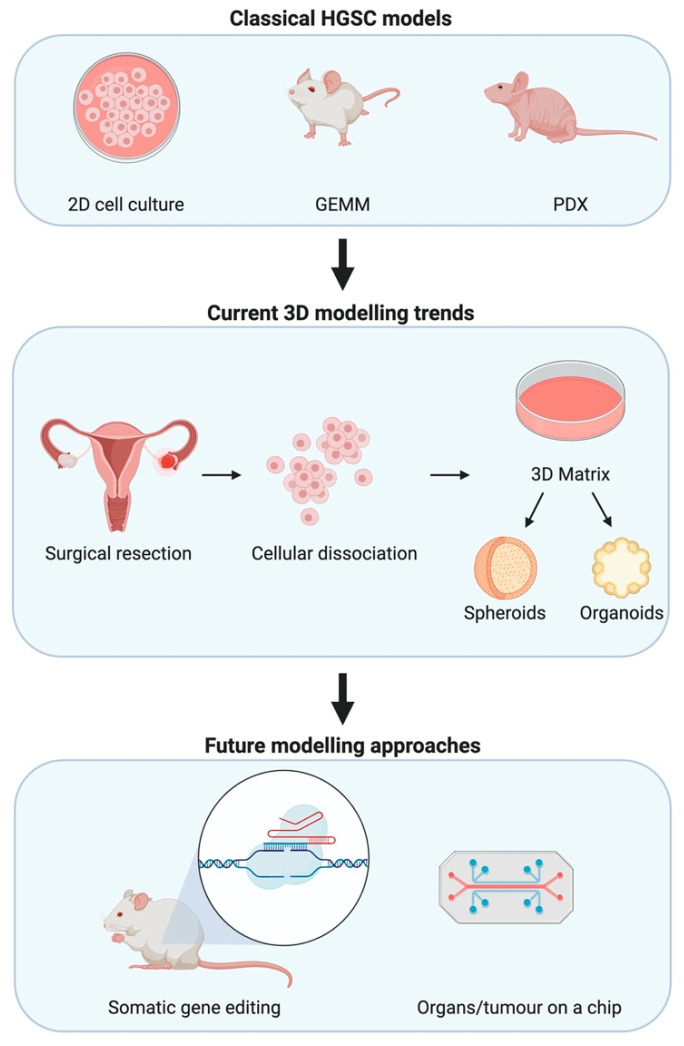
Modelling high-grade serous epithelial ovarian cancer (HGSC) for research. Classical methods to investigate HGSC have depended on 2D mono- and co-cultures, GEMMs, and patient-derived xenografts (PDXs). Advances in in-vitro modelling lead to trends in 3D modelling wherein primary cells are used to form spheroids and organoids that aim to recapitulate interactions between cell types. Whilst technological advances seek to join such 3D modelling advances with microfluidics allowing for organs and tumours on a chip. Somatic gene editing facilitated by advances in CRISPR/Cas9 biotechnology allows for faster oncogenic mutations that are more representative of the real-world scenario.

**Table 1 ijms-21-04806-t001:** The subtypes of Epithelial Ovarian Cancer in women, their frequencies and molecular features and the associated genetically engineered mouse models (GEMMs).

Subtype	Frequency	Molecular Features	GEMMs
Mucinous	<5%	*CDKN2A* copy number alterations [12] *KRAS*, *TP53*, *RNF43*, BRAF, *PIK3CA*, *ARID1A* [12,13] mutations *ERBB2* [12], *HER2* [14], *HOXD9* [15] amplification	*Amhr2-Cre; LSL-Kras^G12D/+^Pten^loxP/loxP^*; Trp53^R172H/+^ [16]
Clear cell	~10%	*ARID1A* [17] and *PIK3CA* [18] mutations Ubiquitous *HNF1β* expression [19] Loss of *PTEN* expression [20]	No GEMMs
Endometrioid	~10%	*ARID1a* [17] and *PPP2R1A* [21] mutations *PTEN* alterations [22]	(ADCRE) *- LSL-K-ras^G12D/+^Pten^loxP/loxP^* [23] (ADCRE) *- Apc^loxP/loxP^;Pten^loxP/loxP^* [24] *Ovgp1-iCre^ERT2^;Apc^loxP/loxP^; Pten^loxP^* [25]
Low-grade serous	<5%	*KRAS, BRAF, ERBB2* mutations [26,27]	*Amhr2-Cre; LSL-Kras^G12D/+^; Pten^loxP/loxP^* [28]
High-grade serous	~70%	*TP53* [29,30] and *BRCA1/2* [30,31] mutations *CCNE1* and *RB1* [32] aberrations	Older GEMMs reviewed in detail (Howell) Most recent HGSC GEMMS: *Pax8-rtTA; TetO-Cre; Brca1^loxP/loxP^; Trp53^mut^; Pten^loxP/loxP^* [33] *Ovgp1-iCreER^T2^; Brca1^loxP/loxP^; Trp53^loxP/loxP^; Rb1^loxP/loxP^; Nf1 ^loxP/loxP^* [34] *Lgr5-Cre; Trp53^R172H/+^; T121* [35]

**Table 2 ijms-21-04806-t002:** Median survival of allograft models.

Syngeneic Cell Line	Engraftment Location	Median Survival (days)
ID8	i.p.	101 *
ID8Trp53^-/-^	i.p.	47 *
ID8Trp53^-/-^;Brca2^-/-^	i.p.	57 *
ID8Trp53^-/-^;Brca1^-/-^	i.p.	46 ^^^
ID8Trp53^-/-^;Pten^-/-^	i.p.	34 ^^^
ID8Trp53^-/-^;Pten^+/-^	i.p.	40.5 ^^^
ID8Trp53^-/-^;Nf1^-/-^	i.p.	36.5 ^^^
ID8Trp53^-/-^;Brca2^-/-^;Pten^-/-^	i.p.	40 ^^^
60577: p53^-/-^;Brca1^-/-^	i.p.	36 ^#^
30200: p53^-/-^;Brca1^-/-^	i.p.	77 ^#^
HGS1: Pax8-Cre;p53^-/-^;Pten^-/-^;Brca2^-/-^	i.p.	91 ^#^
HGS2: Pax8-Cre;p53^-/-^;Pten^-/-^;Brca2^-/-^	i.p.	80.5 ^#^
HGS3: Pax8-Cre;p53^-/-^;Pten^-/-^;Brca2^+/-^	i.p.	87.5 ^#^
HGS4: Pax8-Cre;p53^-/-^;Pten^-/-^;Brca2^+/-^	i.p.	80.5 ^#^

* = [77], ^^^ = [78], ^#^ = [79].

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
