# Peer review of "Modelling Epithelial Ovarian Cancer in Mice: Classical and Emerging Approaches"

_ijms, 2020, doi:10.3390/ijms21134806_

Round 1

Reviewer 1 Report

This is a very interesting, update and well written review the mouse models of high-grade serous.

The manuscript is well structured and covers in an update way all the recent data published.

Minor comment

Lines 259-261. I think that what it is written is not entirely correct. Indeed to my knowledge, the paper reports only the in vitro data with rucaparib showing much higher drug sensitivity only in vitro, not in vivo. The in vivo data relates only to cisplatin. Please amend the text.

Author Response

Reviewer 1: Lines 259-261. I think that what it is written is not entirely correct. Indeed to my knowledge, the paper reports only the in vitro data with rucaparib showing much higher drug sensitivity only in vitro, not in vivo. The in vivo data relates only to cisplatin. Please amend the text.

Thank you for bringing this to our attention. We have revisited lines 259-261 and amended them to read:

“Mutants containing edits to Brca1/2 were significantly more sensitive to PARP inhibition with rucaparib treatment in vitro than those with single mutations in Trp53 only.”

Reviewer 2 Report

The manuscript titled “Modelling epithelial ovarian cancer in mice: classical and emerging approaches” reviewed the recent advancements and utility of HGSC genetically engineered mouse models (GEMMs) to date with further analysis and critique on alternative approaches to modeling HGSC considers technological advancements in somatic gene editing and modeling prototypic organs, capable of tumorigenesis, on a chip for establishing preclinical models with high fidelity to the molecular features of HGSC, the most aggressive subtype of epithelial ovarian cancer.

The manuscript is a well and logically constructed paper; the reviewer still has some concerns.

Comments and suggestion

  1. In figure 1 legends, please align the text of legend.
  2. In Table 2. The median survival of allograft models. The ID8 cell line was derived from explanted ovaries of C57Bl/6 mice, from which OSE was grown in vitro in the presence of EGF and is the most widely used and recognized syngeneic mouse OC cell line. The median survival days of ID8 are 101 days. The tumor behavior of syngeneic cell lines (ID8Trp53-/-, ID8Trp53-/-; Brca2-/-, ID8Trp53-/-; Pten-/-…….) could more closely resemble ovarian cancers. Why are the survival days all shorter than the ID8 cell line? The author should discuss this point in the manuscript.

In my opinion, organoids are currently modeling trends and providing potential alternatives for their use in preclinical research. The author should describe more details from surgical resection to cell dissociation, 3D Matrix, then organoid. The same manner also in the somatic gene editing and tumor-on-a-chip.

Author Response

Reviewer 2 Comment 1: In figure 1 legends, please align the text of legend.

Thank you for bringing this to our attention. The figure 1 legend has now been formatted to ‘justified’.

Reviewer 2 Comment 2: In Table 2. The median survival of allograft models. The ID8 cell line was derived from explanted ovaries of C57Bl/6 mice, from which OSE was grown in vitro in the presence of EGF and is the most widely used and recognized syngeneic mouse OC cell line. The median survival days of ID8 are 101 days. The tumor behavior of syngeneic cell lines (ID8Trp53-/-, ID8Trp53-/-; Brca2-/-, ID8Trp53-/-; Pten-/-…….) could more closely resemble ovarian cancers. Why are the survival days all shorter than the ID8 cell line? The author should discuss this point in the manuscript.

Thank you for your comment. Lines 229 – 238 discuss the differences between the native ID8 cell line and the CRISPR-Cas9 engineered constructs. We have amended (underlined) lines 236-239 to read as noted below, to further emphasize the common difference contributing to shorter survival days:

“It was shown that the Trp53-/- and Trp53-/-;Brca2-/- models had significantly shorter median survival than the native ID8 orthotopic model when grown i.p. in mice, with the single knockout having the shortest median survival of the two (Table 2), demonstrating the importance of Trp53 knockout in survival latency.”

Reviewer 2 Comment 3: In my opinion, organoids are currently modeling trends and providing potential alternatives for their use in preclinical research. The author should describe more details from surgical resection to cell dissociation, 3D Matrix, then organoid (299-301). The same manner also in the somatic gene editing (337-338, 342-343) and tumor-on-a-chip (363-365).

Thank you for your comment. We have amended lines 299-301, 337-338, 342-343, and 363-365 to include more detail on these techniques in a manner that doesn’t detract from the primary focus of the review.